# The C-terminal tail of CSNAP attenuates the CSN complex

Maria G Füzesi-Levi[1], Gili Ben-Nissan[1], Dina Listov[1], Yael Fridmann Sirkis[2], Zvi Hayouka[3], Sarel Fleishman[1], Michal Sharon[1]

Protein degradation is one of the essential mechanisms that enables reshaping of the proteome landscape in response to various stimuli. The largest E3 ubiquitin ligase family that targets proteins to degradation by catalyzing ubiquitination is the cullin–RING ligases (CRLs). Many of the proteins that are regulated by CRLs are central to tumorigenesis and tumor progression, and dysregulation of the CRL family is frequently associated with cancer. The CRL family comprises ~300 complexes, all of which are regulated by the COP9 signalosome complex (CSN). Therefore, CSN is considered an attractive target for therapeutic intervention. Research efforts for targeted CSN inhibition have been directed towards inhibition of the complex enzymatic subunit, CSN5. Here, we have taken a fresh approach focusing on CSNAP, the smallest CSN subunit. Our results show that the C-terminal region of CSNAP is tightly packed within the CSN complex, in a groove formed by CSN3 and CSN8. We show that a 16 amino acid C-terminal peptide, derived from this CSN-interacting region, can displace the endogenous CSNAP subunit from the complex. This, in turn, leads to a CSNAP null phenotype that attenuates CSN activity and consequently CRLs function. Overall, our findings emphasize the potential of a CSNAP-based peptide for CSN inhibition as a new therapeutic avenue.

## Introduction

All cells depend on a balanced and functional proteome, that is, proteostasis, which enables adaptation to external and internal perturbations (1). This highly sophisticated, interconnected system involves protein synthesis, folding and degradation. The functionality of the proteostasis system requires high specificity, as different cell types that are structurally and functionally diverse exhibit distinct proteomes (2). In the case of protein degradation, specificity is mainly achieved by E3 ligases that ubiquitinate distinct proteins, resulting in their degradation (3).

Eukaryotic cells express hundreds of ubiquitin E3 ligases, which can operate in different cellular contexts, respond to numerous cellular signals, and process diverse protein substrates (4). One of the largest E3 ubiquitin ligase families, that account for nearly half of all E3 ligases and is responsible for ubiquitination of 20% of the proteins degraded by the 26S proteasome, comprises cullin–RING ligases (CRLs) (5). CRLs are modular protein complexes that are assembled around a central cullin scaffold, which is associated with a specific substrate receptor, adaptor protein, and a RING protein that recruits the E2 enzyme (6). The nine human cullins and their interchangeable set of substrate specificity factors and adaptors generate about ~300 different configurations of CRL complexes. These complexes coordinate the levels of numerous proteins affecting every facet of eukaryotic cellular processes, such as cell cycle, cellular proliferation, hypoxia signaling, reactive oxygen species clearance, and DNA repair. Remarkably, in spite of the great diversity of CRLs in terms of composition and function, all of them are regulated by the COP9 signalosome complex (CSN) (7).

The 9-subunit CSN complex regulates CRLs by a dual-inhibitory mechanism involving catalytic and non-catalytic functions. The first involves the catalytic subunit CSN5 that enzymatically deconjugates the ubiquitin-like protein Nedd8 from the cullin subunit (deneddylation) (8). The deneddylation of CRLs promotes new substrate binding and prevents autoubiquitination, which may lead to degradation of the substrate recognition subunits of the CRL. The second mechanism is mediated through physical binding to CRLs, sterically precluding interactions with E2 enzymes and ubiquitination of substrates (9, 10). Together, these two mechanisms control the dynamic assembly and disassembly cycles of CRLs that enable adaptation of the complex configuration to the cellular needs.

As a direct regulator of CRLs, the CSN is considered an attractive drug target. Major efforts towards this direction were focused on CSN5, the catalytic subunit of the complex. For example, methods for silencing *CSN5* gene expression were established (11, 12, 13). Moreover, various screening assays were developed for identifying compounds that specifically decrease the deneddylation activity of CSN5 (14, 15, 16, 17). As a result, compounds that inhibit the deneddylase activity of CSN5 and in turn inhibit tumor cell growth have been identified, as doxycycline, an inexpensive, commonly used and well-tolerated antimicrobial agent (18) and the zinc-binding inhibitor CSN5i-3 (18, 19, 20, 21, 22). So far, however, this has not resulted in CSN5 inhibitors that are ready for clinical use.

[1]Department of Biomolecular Sciences, Weizmann Institute of Science, Rehovot, Israel  [2]Life Sciences Core Facilities, Weizmann Institute of Science, Rehovot, Israel  [3]Institute of Biochemistry, Food Science and Nutrition, The Robert H Smith Faculty of Agriculture, Food and Environment, The Hebrew University of Jerusalem, Rehovot, Israel

Correspondence: michal.sharon@weizmann.ac.il

Our discovery of CSNAP, the ninth integral subunit of the CSN complex, led us to propose this subunit as a new therapeutic avenue (23, 24). CSNAP, which exists in a one-to-one stoichiometry with the other CSN subunits, consists of only 57 amino acids (molecular weight: 6.2 kD) (Fig 1A). Its structural characteristics indicate mostly random-coil with transiently formed extended structure in the C-terminus (25). Moreover, cross-linking mass spectrometry (MS) indicates that CSNAP binds the CSN in a cleft formed by CSN1, CSN3, and CSN8, resulting in local subunit reorientations (26), and it was shown that the C-terminal region is essential for its integration into the complex (24). In addition, data indicate that removing CSNAP affects the non-catalytic activity of the CSN complex, as it increases the affinity of CSN towards CRL, although leaving deneddylation activity unchanged (23). This in turn, influences the dynamic plasticity of CRL configuration. Consequently, the absence of CSNAP (CSN$^{\Delta CSNAP}$) alters cell cycle progression and reduces cellular viability. Taking together these observations, we hypothesized that by preventing CSNAP integration within the CSN complex, that is, mimicking the CSN$^{\Delta CSNAP}$ characteristics, the functionality of the complex could be inhibited, impairing cell cycle progression, proliferation, and the adaptive response to oncogenic stress conditions (23).

Here, we show that the cellular presence of a C-terminal CSNAP (C-CSNAP) peptide, prevents the incorporation of the endogenous CSNAP subunit into the complex. Preventing CSNAP association leads to reduced cellular proliferation, as occurs in the CSNAP null phenotype harboring attenuated CSN function (24). Thus, the results open up a new avenue for CSN inhibition. Because an experimentally determined structure for CSNAP in complex with CSN is not yet known, our strategy was based on generating an AlphaFold2 model and challenging it with partial structural and cross-linking data (27). Based on the predicted structure, we selected the C-CSNAP peptide and screened for its optimal properties using peptide array analysis. This approach unravels various properties required from the inhibitory peptide and further supports the predicted CSNAP-bound CSN structure. Overall, our results suggest that preventing CSNAP integration within the CSN complex by an inhibitory peptide opens a new therapeutic avenue for CSN inhibition.

## Results

### The C-terminal tail of CSNAP forms the main interaction region with the CSN complex

In solution, CSNAP is an intrinsically unstructured protein, with a slight tendency to form an extended structure in its C-terminus (25). Within the context of the intact complex, however, the conformation of this subunit is unknown. To assess the CSNAP-bound state, we predicted its structure using AlphaFold2 (AF), which has recently demonstrated atomic-level accuracy in ab initio prediction of protein complexes (28 Preprint, 29). Considering a recent cross-linking MS study that showed that CSNAP is positioned in a cleft formed among CSN1, CSN3, and CSN8 (26), we generated a model of these three subunits together with CSNAP (Fig 1B). The predicted AF

four subunit structure was superimposed onto the X-ray crystallographic structure of the CSN$^{\Delta CSNAP}$ structure (PDB 4D10) (27), showing significant similarity with an RMSD of 1.02 and 0.62 Å for CSN1 and CSN3/CSN8, respectively, and 2.3 Å for the three subunits together. We further evaluated prediction quality based on the AF confidence parameters of ipTM, which is a measure of overall predicted model accuracy weighted more heavily at the oligomeric interfaces, and predicted aligned error which describes the confidence in the relative orientation of the monomers. The predicted four subunit structure displayed a convergent interaction interface with ipTM scores of 0.83 and favorable predicted aligned error values along CSN1, CSN3, CSN8, and the CSNAP C-terminal interfaces (Fig S1A and B). Judging by the low per-residue estimate of AF confidence (pLDTT scores) of the CSNAP's N-terminal region, we estimate that this region is unstructured, and we have omitted it from the structure. This is consistent with our previous results indicating that deletion of the N-terminal region of CSNAP, involving the first 20 amino acids, has no effect on CSNAP integration into the CSN complex (24). Additional support for the importance of the C-terminal region for forming the CSNAP-bound CSN complex comes from the cross-linking MS study in which 14 out of the 16 CSNAP containing cross-links were positioned at the C-terminal region of CSNAP (26).

The model further supports our previous finding that Phe44 and Phe51 are necessary mediators of CSNAP interaction with the CSN (24) (Fig 1C). These aromatic residues are buried within the CSN3 α-helix bundle forming the CSNAP/CSN interface. Moreover, as suggested (24), an amphipathic configuration is formed, in which the aromatic residues face the core of the complex and the negatively charged residues are exposed to the solvent (Fig 1D). We then mapped the previously determined cross-linking constraints (26) of C-CSNAP on the generated model. Linkages involving CSN3 satisfied all eight constraints, whereas positions 472 and 477 on the C-terminus of CSN1 receive low pLDTT scores and are predicted to be unstructured, thus precluding comparison. The distance between CSN3 Glu133 and CSNAP Asp42, Asp46, Asp48, and Asp49 was below 25 Å, and likewise, the distance between CSN3 Glu333 and CSNAP Asp52, Asp53, Asp54, Asp55 was below 20 Å. Taken together, the results indicate that C-CSNAP forms the main binding interface with the CSN complex, leading us to suggest that a synthetic C-terminal peptide that will accommodate this region, may preclude the integration of CSNAP into the complex, thereby generating the CSN$^{\Delta CSNAP}$ phenotype wherein the complex function is inhibited.

### C-CSNAP peptides displace CSNAP from the CSN complex

To define whether C-CSNAP can displace the endogenous CSN subunit from the complex, we transiently expressed several versions of C-CSNAP peptides with differing lengths, fused to the fluorescent protein cerulean (Cer) in WT HAP1 cells (Fig 2A) and cells lacking CSNAP (ΔCSNAP cells) (Fig S2A). We then performed co-immunoprecipitation analyses, using an antibody against GFP that recognizes the Cer tag (Figs 2B and C and S2B). The results indicated that in WT cells, C-CSNAP is pulled down, together with CSN5, indicating its integration within the complex. Moreover, in ΔCSNAP cells, all C-CSNAP derivatives were efficiently incorporated into CSN

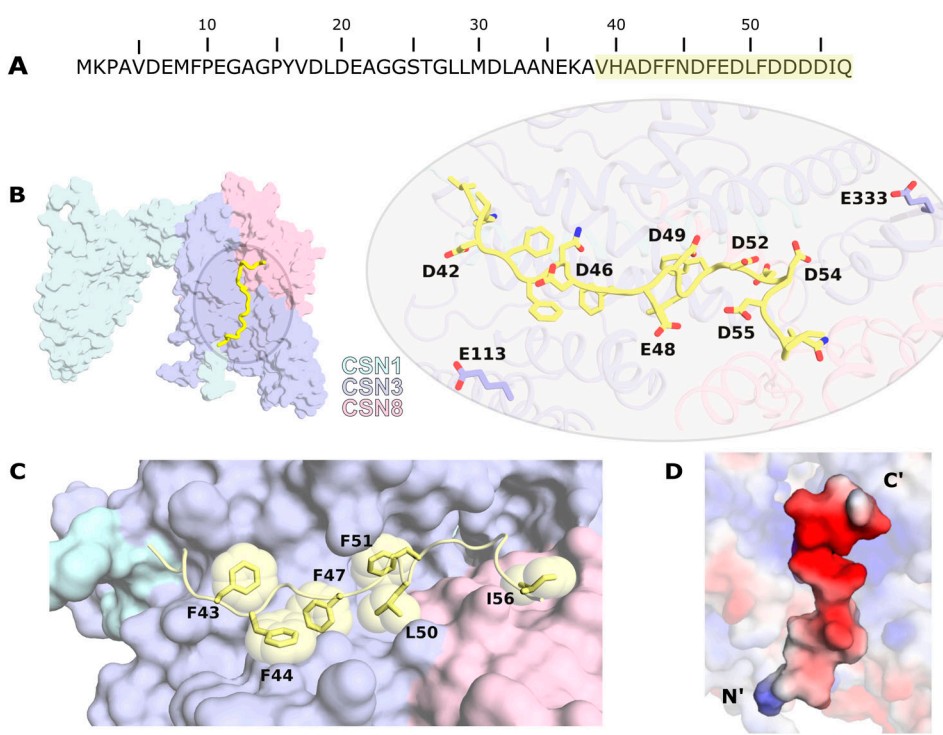

**Figure 1. CSNAP C-terminus binds within a groove formed by CSN3 and CSN8.**
**(A)** The sequence of CSNAP. The C-terminal region of CSNAP is highlighted in yellow. **(B)** AlphaFold-Multimer prediction of the structure of CSN1, CSN3, CSN8, and the C-terminal region of CSNAP. The surfaces of CSN1, CSN3, and CSN8 are colored in cyan, purple, and pink, respectively. The C-terminal region of CSNAP, which displayed high AF confidence, is in inset close-up represented by yellow cartoon. Polar and charged residues are shown in sticks. CSN3 E113 and E333 were used for cross-linking distance analysis. **(C)** Close-up of CSNAP's-binding region. The hydrophobic and aromatic residues of CSNAP (yellow spheres) are largely buried within the groove formed by CSN3 and CSN8. **(D)** Electrostatic potential distribution maps of CSN3, CSN8, and CSNAP colored by charge (red negative and blue positive). Maps for CSN3/CSN8 and CSNAP were generated separately. The image emphasizes the negative charge of CSNAP and the complementary positive potential of the CSN3/CSN8 groove.

(Fig S2B), whereas in WT cells, differential integration was observed (Fig 2B). The 16 amino acid fragment of C-CSNAP–Cer displayed the highest degree of association with the CSN complex. Surprisingly, this peptide outperformed the full-length CSNAP–Cer protein in its efficiency in displacing the endogenous CSNAP subunit. Truncations of the 16 amino acid stretch from either end decreased or abolished the incorporation efficiency (Fig 2B and C).

## Optimization of the C-CSNAP peptide length and sequence

To generate an optimized C-CSNAP peptide, we used the peptide array methodology. The array was designed to screen for the optimal peptide length, sequence, and stability (Fig 3A). A recombinant CSN complex was expressed, purified, and screened for binding to the peptide array by using an anti-CSN3 antibody. To rule out non-specific binding of the probing antibody, a control experiment was carried out without the addition of the CSN complex (Fig 3A right panel). The anti-CSN3 antibody bound to one spot, which was disregarded for further analysis. The intensity of each spot, reflecting the interaction strength, was normalized to the 16 amino acid C-terminal fragment of CSNAP (Fig 3B–G and Table S1).

Sequential elongation of the C-CSNAP peptide towards the N-terminal enhanced the binding to CSN, with the addition of four amino acids showing the highest signal (Fig 3B). This is likely due to the polar interaction, that is formed in this peptide between $^{CSNAP}$His41 and $^{CSN3}$Ser127, as revealed by the predicted structure (Fig 1). Furthermore, alanine scanning, in which each residue in C-CSNAP was substituted to alanine, was performed to determine the residue-specific contribution to the CSN interaction interface. The assay showed that substitution of the negatively charged

(Asp42 and Asp46) and hydrophobic residues (Phe47, Leu50, and Phe51) reduced the binding to the complex. These results correlate with the model (Fig 1B–D), as the aspartic acid residues are solvent exposed and hence disfavor replacement to a non-polar residue, whereas the bulky hydrophobic residues are buried within the CSN3 interface and their substitution to alanine forms cavities. We also examined the impact of conservative substitutions by replacing the aromatic residue phenylalanine to tryptophan and the negatively charged aspartic acid residue to glutamic acid, in attempts to improve binding (Fig 3D and E). Decreased tolerance to substitution was detected upon replacement of phenylalanine to the bulkier and less hydrophobic tryptophan residue, supporting the notion of tight packing of the four tryptophans in C-CSNAP (Fig 1D). The exchange of Asp to Glu at position 42 (D42E), however, increased the binding, suggesting that the larger solvent exposed polar surface of glutamic acid eases the formation of hydrogen bonds with the aqueous solvent (Fig 1E).

Next, to increase the analyzed chemical space, we examined the ability of peptides bearing non-canonical amino acids, that is, D-amino acids and the helix-promoting α-aminoisobutyric acid (Aib), to bind CSN. Peptide sequences containing D-amino acids are attractive for our use as they are highly resistant to degradation by proteases (30, 31). However, the impact of such replacement may interfere with protein interaction, and hence we examined the effect of a single L-amino acid replacement to D-amino acid on the binding capacity to CSN. The results pointed to the importance of Phe47 and Glu48 in CSNAP–CSN interaction, as their substitution reduced significantly the binding of the peptides (Fig 3F). On the other hand, D-amino acid substitution of Leu50 to Asp55 exhibited enhanced CSN binding, by more than 20%, in comparison to the

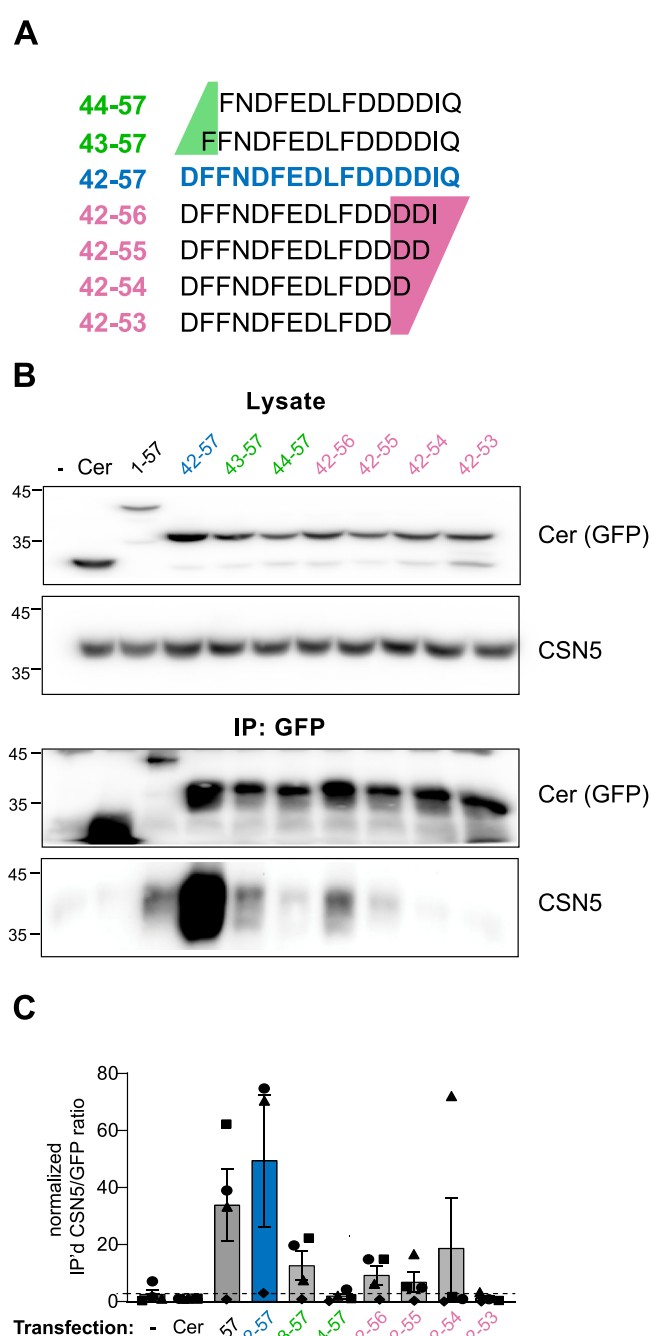

## C

**Figure 2. The last 16 amino acids of CSNAP efficiently displace the endogenous CSNAP subunit.**

**(A)** The different CSNAP C-terminal sequences that were examined. The 16 amino acid sequence is highlighted in blue, and the N- or C-terminally truncated forms are labelled in green and magenta, respectively (numbering of residues are corresponding to full length CSNAP). **(B)** Immunoprecipitation experiments of cells transiently expressing full-length CSNAP (black) or truncated versions (green, blue, and magenta) fused to cerulean (Cer). Lysates from HAP1 cells transiently expressing CSNAP variants are shown on the top, and the corresponding anti-GFP pull downs on the bottom. The last 16 amino acids of C-CSNAP fused to Cer (blue) were preferentially incorporated into the COP9 signalosome complex. The efficiency of incorporation of the 16 amino acid C-CSNAP outperforms even that of the full-length CSNAP subunit. **(C)** Quantification of the densitometry results of at least three independent immunoprecipitation experiments using an anti-GFP antibody, represented as mean ratios of immunoprecipitated CSN5/GFP, followed by normalization to the sample transfected with Cer, plotted as mean ± SEM.

original peptide. Single-residue D-amino acid substitutions are known to disrupt $\alpha$-helical structure (30) and given the intrinsically unstructured conformation of C-CSNAP (Fig 1), such replacements are probably stabilizing the unstructured conformation of this region, suggesting that such partial D-amino acid substitution may be useful for in vivo activity and stability. Incorporation of an Aib residue, in which the $\alpha$-hydrogen atom of alanine is replaced with a methyl substituent is commonly used for promoting helicity (32). This substitution did not significantly affect the ability to interact with CSN when introduced at the first two positions of the peptide but attenuated binding in all internal substitutions (Fig 3G). This observation is in agreement with the impact of the D-amino acid substitutions, highlighting the prevalence of an unstructured conformation of C-CSNAP. Lastly, N-terminal methylation of the peptide, which improves pharmacokinetic properties (33), did not affect the interaction with CSN (Fig 3G).

The same array was also reacted with the CSN$^{\Delta CSNAP}$ complex (Fig S3A–D and Table S2). The data indicate that the interaction of CSN with C-CSNAP peptides is more stringent than CSN$^{\Delta CSNAP}$, as expected from the requirement to expel the endogenous CSNAP subunit from the complex before peptide binding. Taken together, the peptide array analysis not only provided insight into the optimal C-CSNAP peptide characteristics but also validated and supported the predicted model.

To validate the peptide array results, we used the surface plasmon resonance biosensor (Biacore) method to measure the binding affinities between C-CSNAP peptides and CSN$^{\Delta CSNAP}$. We selected three peptides, the 16 residue C-terminal fragment ($^{42}$DFFNDFEDLFDDDDIQ$^{57}$, 16AA C-CSNAP) (Fig 4B), the 20 amino acid peptide ($^{38}$AVHADFFNDFEDLFDDDDIQ$^{57}$, 20AA C-CSNAP) (Fig 4C), and a peptide containing the substitution of Asp to Glu at position 42 (D42E C-CSNAP) (Fig 4D). As a positive control, we used full-length CSNAP (Fig 4A), and a scrambled peptide (IDGENVSNLDYARKAT) (Fig 4E) was used as a negative control. Interaction between the CSN$^{\Delta CSNAP}$ complex and the CSNAP variants was measured using a setting of 180 s binding followed by 1,000 s dissociation time (Fig 4). The results indicate that the 16 and 20 amino acid C-CSNAP peptides bind the CSN complex with a similar $K_d$ of $4.0 \times 10^{-10}$ and $4.3 \times 10^{-10}$, respectively (Fig 4F). The D42E displayed a slightly lower affinity of $2.5 \times 10^{-9}$, and the highest affinity was for the full-length CSNAP protein $6.8 \times 10^{-11}$. No measurable binding was detected for the scrambled peptide. Overall, the data show the specificity and high affinity of the CSN$^{\Delta CSNAP}$ complex to CSNAP and C-CSNAP variants.

## C-CSNAP peptides substitute the endogenous subunits

We next focused on exploring the ability of C-CSNAP peptides to displace the endogenous CSNAP subunit, which like the other CSN subunits is present in equimolar amounts. To this end, we selected four peptides, the 16 and 20 amino acid peptides (16AA and 20AA C-CSNAP, respectively), a peptide containing the substitution of Asp to Glu at position 42 (D42E C-CSNAP), and a peptide in which Leu50 was exchanged with a D-amino acid (dLeu50 C-CSNAP) (Fig 5). The CSN complex was incubated with each one of the peptides, bound to StrepTactin beads (StrepIIx–CSN3) and the unbound CSNAP fraction was removed by washing (Fig 5A). Next, the complex was

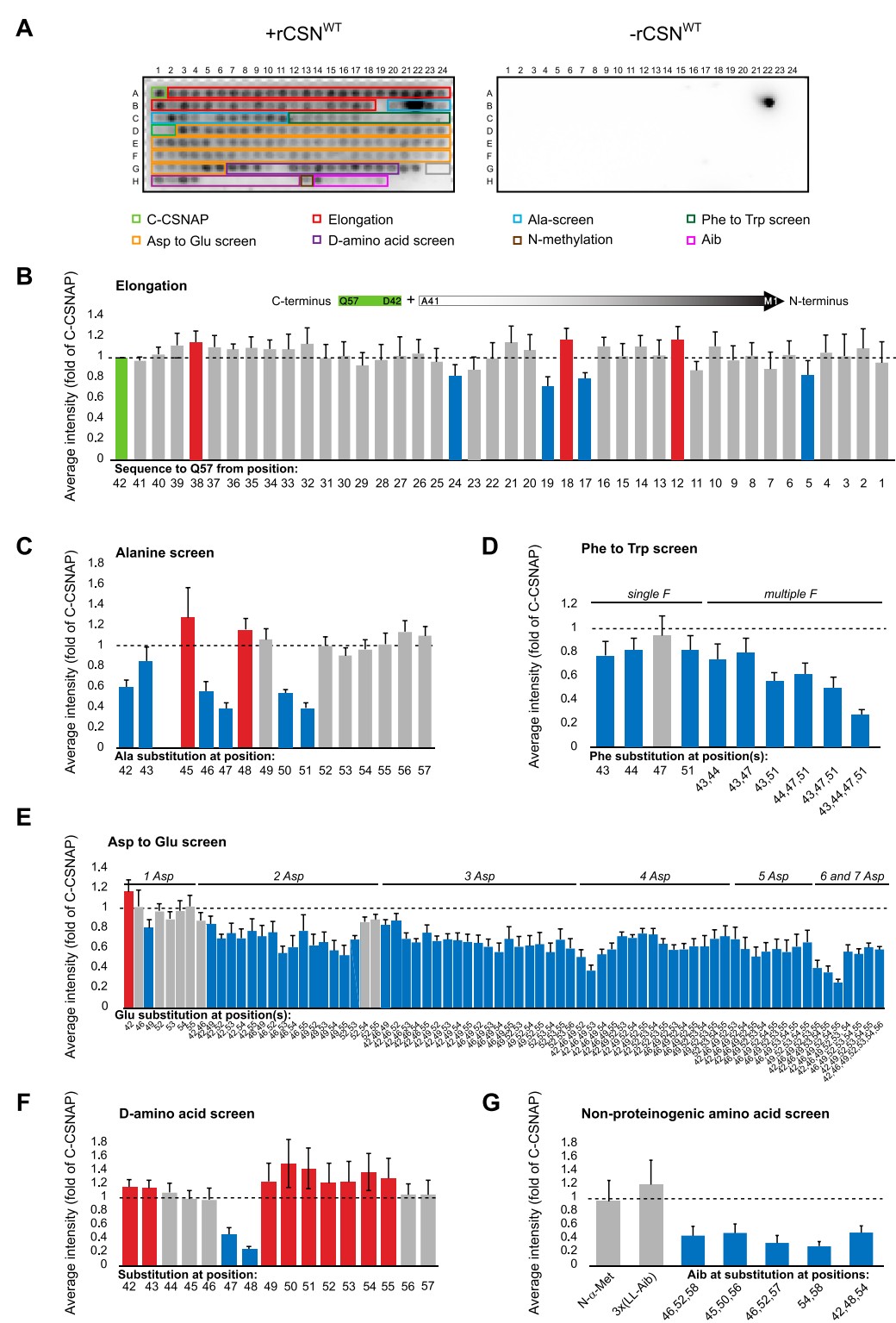

**Figure 3. Peptide array screen of modified C-CSNAP peptides.**
C-CSNAP peptide and its derivatives were synthesized on a microarray. The array was blocked and incubated with or without the COP9 signalosome complex (CSN) and probed using an anti-CSN3 antibody and HRP-conjugated secondary antibody. To assess non-specific binding of the antibodies to the array, anti-CSN3 and anti-GAR–HRP were used without the addition of the CSN. Signal intensity of each spot was measured and normalized to the intensity of the spot of C-CSNAP (A1). Measured values for each spot were averaged from six arrays and plotted with SEM. Red bars indicate increased binding compared with C-CSNAP (>1.14-fold, [average – SEM] > 1), and blue bars represent weaker interaction (<–1.14-fold, [average + SEM] < 1), all other bars are shown in gray. In (C), results for spot B22 was disregarded because of non-specific CSN binding to this spot. **(A)** A representative image of the peptide array (left) and the background (right). **(B)** Elongation of the C-terminal 16 amino acids (green) with one residue at a time based on the full-length sequence. The data highlight stronger binding to CSN when the sequence is 20 amino acids long. **(C)** The bar plot shows the

separated into its constituent subunits using a monolithic column ([34](ref)). The eluted subunits were sprayed directly into the mass spectrometer for intact protein mass measurements ([Fig 5B](fig)). As a negative control, we used the CSN complex without the addition of C-CSNAP. Analysis of the spectra validated the presence of charge series corresponding in mass to the four C-CSNAP peptides, along with the other CSN subunits, indicating that all designed peptides were able to displace the endogenous CSNAP subunit ([Figs 5](fig) and [S4](fig)). However, the 20AA C-CSNAP peptide displayed a higher capacity for dislocating CSNAP, as shown by its increased abundance relative to the other peptides ([Fig 5C](fig)). The incorporation of 20AA C-CSNAP within the CSN complex was also validated by native-MS/MS analysis ([Fig 5D](fig)).

### C-CSNAP peptides reduce cellular proliferation

To clarify the ability of C-CSNAP peptides to displace CSNAP in cells, we stably expressed four versions of CSNAP, the 16AA and 20AA C-CSNAP, D42E and scrambled peptides fused to the fluorescent protein Cer in HAP1 cells. As control, we used HAP1 cells expressing Cer only. We then performed reciprocal co-immunoprecipitation analyses, using antibodies against CSN3 and GFP recognizing Cer ([Figs 6A](fig) and [S5](fig)). The results indicated that the 16, 20AA, and D42E C-CSNAP peptides are integrated into the CSN complex in a different efficiency. Based on band intensities, the longer peptide, 20AA C-CSNAP, displayed the highest ability to integrate into CSN, and the efficiency was lower for the 16AA peptide and lowest in D42E. The scrambled peptide failed to incorporate into the CSN complex, indicating the specificity of the C-CSNAP peptides with the CSN. This is reflected by the higher intensity of the GFP band upon CSN3 pull down, and by the more intense CSN3 band following GFP co-immunoprecipitation.

Previously, we have shown that ΔCSNAP cells proliferate at a slower rate compared with WT cells ([23](ref), [24](ref)). We hypothesized that displacement of the endogenous CSNAP by the exogenously expressed C-CSNAP peptide fused to Cer will induce a similar effect, that is, reducing cell proliferation, similar to ΔCSNAP cells. We, therefore, compared the proliferation rates of WT, 16AA, and 20AA C-CSNAP–Cer expressing cell lines in resazurin-based viability assays. As a control, we used a cell line expressing only Cer. The data show that there are significant reductions in cell proliferation in both 16AA and 20AA C-CSNAP clones compared with the cells expressing only Cer ([Fig 6B](fig)). In summary, this result confirmed the cellular ability of a C-CSNAP–derived peptide to dislocate the endogenous subunit and thus attenuate CSN activity.

## Discussion

Here, we showed that a C-terminal peptide of CSNAP can displace the endogenous CSNAP subunit. We demonstrated that the peptide is incorporated within the complex, and that upon its binding, cell proliferation is reduced, thereby, mimicking the phenotype observed upon deletion of CSNAP ([24](ref)). The results are based on an AlphaFold2-generated high-confidence prediction model of CSN1, CSN3, CSN8, and CSNAP, that agrees very well with our mutational analysis ([24](ref)), peptide array results (current study), with the previously solved X-ray CSN$^{\Delta CSNAP}$ structure ([27](ref)), and chemical cross-linking constraints ([26](ref)). The model provides important insight into the structural details of CSN-bound CSNAP. Coupling the structural and experimental results revealed that a 16 amino acid residue peptide displays the highest cellular capacity for substituting the full-length subunit from the CSN. Overall, our results suggest that inhibition of the CSN complex can be achieved by a peptide-based approach.

In recent years, pharmaceutical research is revisiting the use of peptides as therapeutics, especially for inhibiting protein–protein interactions that are more difficult to therapeutically target than the interactions of globular proteins with other biomolecules ([35](ref), [36](ref), [37](ref)). This is because of the fact that peptide-based drugs benefit from high target specificity, strong binding affinity, low immunogenicity, a lower potential for drug–drug interaction, and high tolerability and safety, whereas effective therapeutic outcomes can be reached with only a small concentration of peptide ([38](ref), [39](ref)). Moreover, advances in structural biology, recombinant biologics, and new synthetic and analytic technologies have significantly accelerated peptide drug development ([40](ref)). There are still major challenges to overcome before wide clinical application of peptides can be considered. Susceptibility to proteolytic degradation, delivery, and rapid renal clearance are known limitations of peptide-based therapies ([41](ref)). Despite these challenges, more than 80 therapeutic peptides have reached the global market to date, and over 170 peptides are in active clinical development, with many more in preclinical studies ([40](ref)), emphasizing the prospective of our approach.

IDPs often fold into ordered secondary folds upon binding to their physiological interaction partners ([42](ref)). The C-terminal region of CSNAP, however, remains extended in its CSN-bound state, as predicted by the model and verified by experimental input. Although less common, multiple examples of IDPs remaining extended upon binding exist ([43](ref)). For instance, the binding of the intrinsically disordered regulatory region of CFTR to NBD1 ([44](ref)), the binding of the IDP Sic to Cdc4 ([45](ref)), or the flexible stretch that remains in the structure of p21 and p27 when bound to CDK ([46](ref), [47](ref)). Even though C-CSNAP does not adopt a secondary structure, it seems to be tightly anchored to CSN3 and CSN8 by its five aromatic residues and complementary electrostatic interactions. Thus, it likely adopts a confined extended conformation rather than populating an ensemble of conformational states. The use of peptides for targeting protein–protein interactions mediated by IDPs is still in its infancy, however, examples of this strategy are already available, such as the interaction of α-synuclein and TPPP/

---

key positions in which single-residue substitution with alanine significantly reduced the binding to CSN. **(D)** Phe substitution screen to Trp demonstrates the specificity of the Phe residues in CSN binding, the more Phe are substituted with Trp the interaction decreased. **(E)** Asp to Glu substitution emphasizes the importance of Asp residues. Single substitution increased slightly the strength of binding to CSN, whereas multiple substitutions significantly weakened interaction in most cases. **(F)** Single D-amino acid substitution at various positions. Substitution of Phe47 significantly reduced binding to CSN, whereas replacement of Leu50 to Asp55 promoted interaction. **(G)** Incorporation of non-proteinogenic amino acids disrupted the CSN–C-CSNAP interaction.

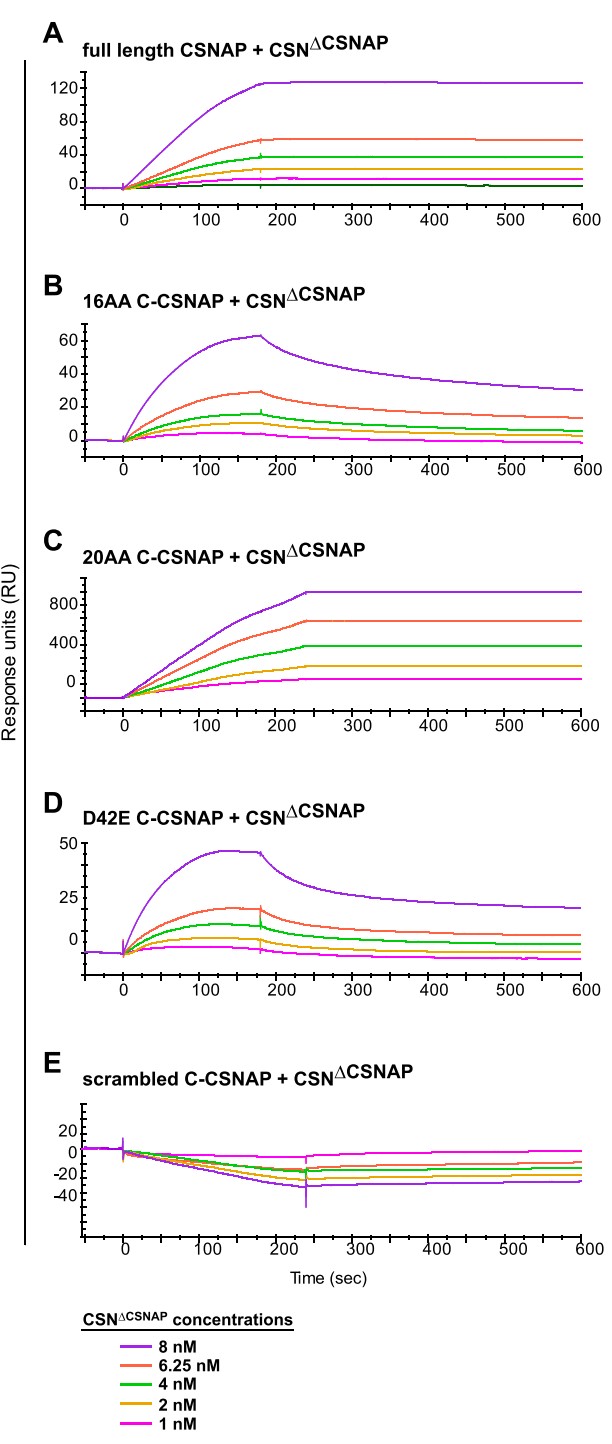

**CSN<sup>ΔCSNAP</sup> concentrations**

| Color | Concentration |
|---|---|
| purple | 8 nM |
| orange | 6.25 nM |
| green | 4 nM |
| yellow/gold | 2 nM |
| pink | 1 nM |

**F**

| | $k_{ON}$ (1/Ms) | $k_{OFF}$ (1/s) | $K_D$ (M) |
|---|---|---|---|
| **16AA C-CSNAP** | $3.5 \times 10^6$ | $1.4 \times 10^{-3}$ | $4.0 \times 10^{-10}$ |
| **20AA C-CSNAP** | $2.5 \times 10^6$ | $1.0 \times 10^{-3}$ | $4.1 \times 10^{-10}$ |
| **D42E C-CSNAP** | $1.9 \times 10^6$ | $2.7 \times 10^{-3}$ | $2.5 \times 10^{-9}$ |
| **full length CSNAP** | $2.4 \times 10^6$ | $1.6 \times 10^{-4}$ | $6.8 \times 10^{-11}$ |
| **scrambled C-CSNAP** | NA | NA | NA |

p25 (48), AF4 and AF9 (49), p53 to MdM2 (50) the interactions of the histones H4 and H2A (51), and NURP1 (52). Such peptides can be used as starting structures for developing peptidomimetics. Methodologies relevant for IDP peptidomimetics are already being developed, possessing protease resistance and increased cell permeability (53), supporting their potential use of this approach for pharmaceutical intervention.

In summary, the main finding of this study was the demonstration that C-CSNAP peptides inhibit CSN activity. The inhibition of CSN is an attractive goal for cancer research because this complex is associated with cancer development and progression. The role of CSN in tumorigenesis is likely indirect and is linked to coordinating CRLs function. Cell division and signal transduction are tightly regulated by CRL activity, thus inhibiting CRLs through down-regulation of CSN is a promising therapeutic avenue. The impact of CSN inhibition on protein degradation is more restricted compared with the proteasome complex, as only the degradation of CRLs substrates is attenuated, possibly leading to reduced resistance and side effects compared with the treatment of proteasome inhibitors. Taken together, the peptide-based strategy we proposed here has the potential to be further developed as lead compounds opening a new avenue for selective CSN inhibition (54, 55).

## Materials and Methods

### AlphaFold model prediction

To implement AlphaFold2 (28 *Preprint*, 29) locally, we used an adapted code written by ColabFold (56 *Preprint*). The run used the five model parameters, seven recycle rounds, and no templates or Amber relaxation. Multiple-sequence alignments were generated through the MMseqs2 API server (57, 58, 59).

### Expression of GFP constructs

The C-terminal <sup>42</sup>DFFNDFEDLFDDDDIQ<sup>57</sup> (C-CSNAP) sequence of CSNAP was generated in pHyg vector coding for full-length CSNAP fused to the N-terminus of Cer through a short Gly–Ser–Gly–Ser linker, by deleting <sup>1</sup>M-<sup>41</sup>A residues of CSNAP using Q5 mutagenesis kit (NEB). N- (-<sup>42</sup>D, -<sup>42</sup>DF) and C-terminal truncations (-<sup>57</sup>Q, -<sup>56</sup>IQ, -<sup>55</sup>DIQ, and -<sup>54</sup>DDIQ) were performed on phyg–C-CSNAP–Cer using the Q5 mutagenesis kit (NEB) (Table S3). HAP1 WT cells were transfected with 5 μg plasmid DNA using jetPRIME (Polyplus), and the culture medium was changed 4 h post-transfection. Stably expressing WT cell

---

**Figure 4. C-CSNAP peptides display high affinity towards the CSN<sup>ΔCSNAP</sup> complex.**
**(A, B, C, D, E)** Biacore sensograms for (A) full-length CSNAP, (B) 16 residue C-terminal fragment, 16AA C-CSNAP, (C) the 20 residue C-terminal fragment, 20 AA C-CSNAP, (D) a peptide containing the substitution of Asp to Glu at position 42, D42E C-CSNAP, and (E) a scrambled peptide. Biotinylated CSNAP variant peptides were immobilized onto a Series S SA sensor chip at a concentration of 2 μg/ml and CSN<sup>ΔCSNAP</sup> was injected at concentrations of 1 (pink), 2 (yellow), 4 (green), 6.25 (orange), and 8 (purple) nM. **(F)** Binding rate constants and affinities were determined via Biacore Evaluation Software analysis of SPR sensograms.

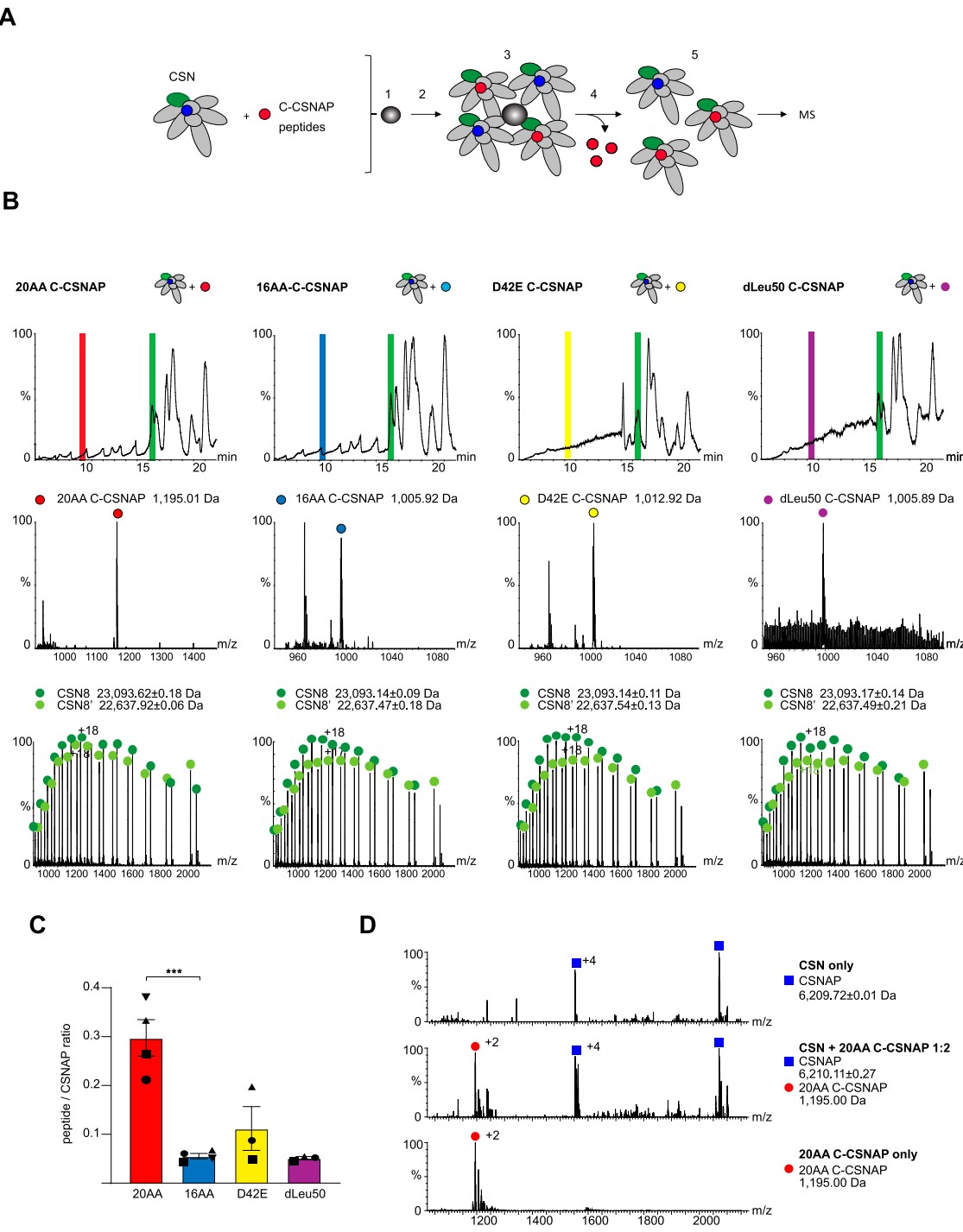

**Figure 5. C-CSNAP peptides displace the full-length subunit.**
**(A)** Schematic representation of the displacement experiments using C-CSNAP peptide variants and recombinant COP9 signalosome complex (CSN). For clarity, CSNAP is colored in blue, C-CSNAP peptides in red, and Csn8 in green. CSN was incubated with 10-fold molar excess of 16AA C-CSNAP or modified peptides before coupling to StrepTactin beads (1) for 2.5 h at 37°C (2). StrepTactin-bound CSN (3) was washed three times (4) to remove free peptides, and CSN was eluted from the beads (5) for mass spectrometry analysis. **(B)** In this analysis, the CSN was separated into its component subunits, using a monolithic column under denaturing conditions. The eluted subunits are directed straight into the mass spectrometer for intact protein mass measurements. Masses corresponding to all four peptides were detected (highlighted with red, light blue, yellow, and dark purple for 20AA, 16AA, D42E, and dLeu50 C–CSNAP, respectively). Representative spectra of the canonical CSN subunit, CSN8 (green). The two different series of peaks in the ESI-MS spectrum correspond to full-length CSN8 and its alternative translation initiation site isoform (34). **(C)** The bar plot shows average ratios of intensities corresponding to masses of each peptide and CSNAP after incubation with CSN with SEM. Significance was calculated from a minimum of three experiments for each peptide using one-way ANOVA, yielding a value of $P < 0.005$, followed by Dunnett's multiple comparisons test ***$P < 0.001$. **(D)** The CSN complex was incubated either with or without the 20AA C-CSNAP peptide at a ratio of 1:2 and then analyzed by tandem MS. MSMS analysis of the intact CSN complex resulted in dissociation of CSNAP (upper panel, dark blue squares). Activation of the CSN complex following preincubation with 20AA C-CSNAP resulted in dissociation of both CSNAP

lines were prepared by applying hygromycin (0.4 mg/ml) selection for 7 d following transfection, then the pool of surviving cells were sorted on a FACSAriaIII cell sorter for Cer-positive cells.

Expression levels were monitored by loading 30 µg of each lysate on 12% Laemmli SDS–PAGE, transferred to PVDF membrane, and probed with anti-GFP antibody (ab290; Abcam).

## Cell lines and expression

Cer, 16AA C-CSNAP–Cer, and 20AA C-CSNAP–Cer expressing stable cell lines were prepared using HAP1 WT cells. Cells were transfected with phyg–Cer, phyg–16AA C-CSNAP–Cer, or phyg–20AA C-CSNAP–Cer, and after 7 d selection with hygromycin, Cer-positive cells were sorted in a BD FACSAriaIII cell sorter. For transient expression of the C-CSNAP constructs, HAP1 WT and ΔCSNAP cells were transiently transfected for 48 h with pHyg–C-CSNAP–Cer, N- or C-terminally truncated forms, or HAP1 WT cells with pTWIST–CMV–hyg–D42E C-CSNAP and pTWIST–CMV–hyg–scrambled C-CSNAP constructs.

## Immunoprecipitation (GFP)

Cells were lysed (50 mM Tris–HCl, pH 7.4, 150 mM NaCl, 0.5% NP-40, 2.5 mM Na-pyrophosphate, 1 mM β-glycero-phosphate, and 1 mM Na-ortho-vanadate supplemented with PMSF, benzamidine, and pepstatin). For each immunoprecipitation, 250 µg lysate was incubated with 1 µl anti-GFP (ab290; Abcam or 2 µl anti-CSN3 ab79698; Abcam) in 500 µl Tris buffered saline (TBS) overnight, then with 20 µl protein-G Sepharose for 1 h, and bound proteins were eluted with 50 µl 2x Laemmli sample buffer.

## BIAcore assays (surface plasmon resonance biosensor assay)

The experiments were carried out using Biacore SA200 SPR sensor (Biacore, GE Healthcare) with control software version 1.1 and Cytiva Series S Sensor chip SA. C-terminally biotinylated peptides, full-length CSNAP, 16AA, 20AA, D42E, and scrambled C-CSNAP peptides were immobilized on a chip, each one in a separate flow cell at 2–5 µg/ml concentration, in a buffer containing 50 mM Tris, pH 7.4, 250 mM NaCl, 5% glycerol, 2 mM EDTA. ~200 and 800 response units (RU) of each peptide was immobilized per flow cell (except cell one for control), on each chip at a comparable level. Recombinant CSN$^{\Delta CSNAP}$ was diluted in the same buffer to 1–8 nM concentration and injected to the instrument at a flow rate of 30 µl/min, allowing 180 s for binding and 1,000 s for dissociation. The chip was regenerated with 10 mM NaOH. All the assays were performed at 25°C. Data were analyzed in SA200 evaluation software version 1.1, with automatic fitting using parameter setting 1:1 binding (ka, kd, and tc global fit, Rmax and RI local fit).

## MS

The monolithic-LC-MS approach applied on the recombinant CSN complex was performed as previously described (34). Briefly, 76.5 pmoles of recombinant CSN$^{WT}$ was incubated with 10-fold molar excess (765 pmoles) of 16AA, 20AA, dLeu50, or D42E C-CSNAP peptides in 50 µl CSN wash buffer (50 mM Tris–HCl, pH 7.4, 250 mM NaCl, 2 mM EDTA, 2 mM DTT) 3 h at 37°C. CSN$^{WT}$ alone or with each peptide, and peptides alone were incubated with 10 µl StrepTactin beads (IBA) in 500 µl at 4°C for 2.5 h. The beads were washed with 200 µl CSN wash buffer, then twice with 200 µl 10 mM HEPES, pH 7.5, and bound CSN was eluted with 30 µl 2.5 mM destiobiotin in 10 mM HEPES, pH 7.5, for 30 min at 37°C. 5 µl of the 30 µl was injected to the monolithic column and denatured with a gradient 10–50% acetonitrile + 0.035% TFA + 0.05% formic acid (20 min). Ratio between the bound peptide (+2) and CSNAP (sum of intensities of charge states +5, +4, +3) was calculated for each sample in three or four replicates, and statistical significance between CSN-bound peptide/CSNAP ratios were compared using one-way ANOVA ($P < 0.005$), followed by Dunnett's multiple comparisons test ($P < 0.001$). For native MS analysis, the CSN complex was buffer exchanged into 250 mM ammonium acetate, incubated either with or without the 20AA C-CSNAP peptide at a molar ratio of 1:2 at 37°C for 3 h, and then analyzed by tandem MS as in reference 60.

## Recombinant CSN expression

Recombinant CSN$^{\Delta CSNAP}$ and CSN was expressed in Sf9 insect cells and purified as described in reference 61.

## Peptide array

Optimization of C-CSNAP sequence for enhanced binding to recombinant CSN$^{\Delta CSNAP}$ was screened by printing the modified sequences of CSNAP or C-CSNAP (Phe to Trp, Asp to Glu, Ala screen, D-amino acid screen, N-α-methylation, and Aib incorporation on INTAVIS Celluspots array [2 × 384spots]). The array was blocked with 5% skim milk in TBS-0.5% Tween 20 (TBS-T) for 4 h at room temperature, washed three times in TBS-T, and then incubated with 0.5 µM CSN$^{\Delta CSNAP}$ in 5% milk–TBS-T. Bound CSN$^{\Delta CSNAP}$ was visualized by incubating with anti-CSN3 (ab79698) antibody (1:10,000, overnight) and goat-anti-rabbit-HRP secondary antibody (1:10,000, 1 h).

## Cell proliferation assays

HAP1 WT cells stably expressing Cer, 16AA, and 20AA C-CSNAP–Cer were trypsinized and counted. Cells from each cell group were plated in a 24-well plate (four replicates, 5,000 cells/well) for proliferation assay. Resazurin-based assays were performed as described in reference 23, 24 h following plating. Fluorescence intensities at 560/600 nm (ex/em) were measured, and average intensities of four replicates in each plate were calculated, normalized against WT after background subtraction, and normalized values from five independent experiments were compared on a bar graph with SEM. Significance was calculated using repeated

and 20AA C-CSNAP (middle panel, dark blue squares, and red circles, respectively), demonstrating that before MSMS analysis, 20AA C-CSNAP was physically associated with the CSN complex. As a control, C-CSNAP was measured alone and was detected as a doubly charged ion (lower panel, red circle).

**A**

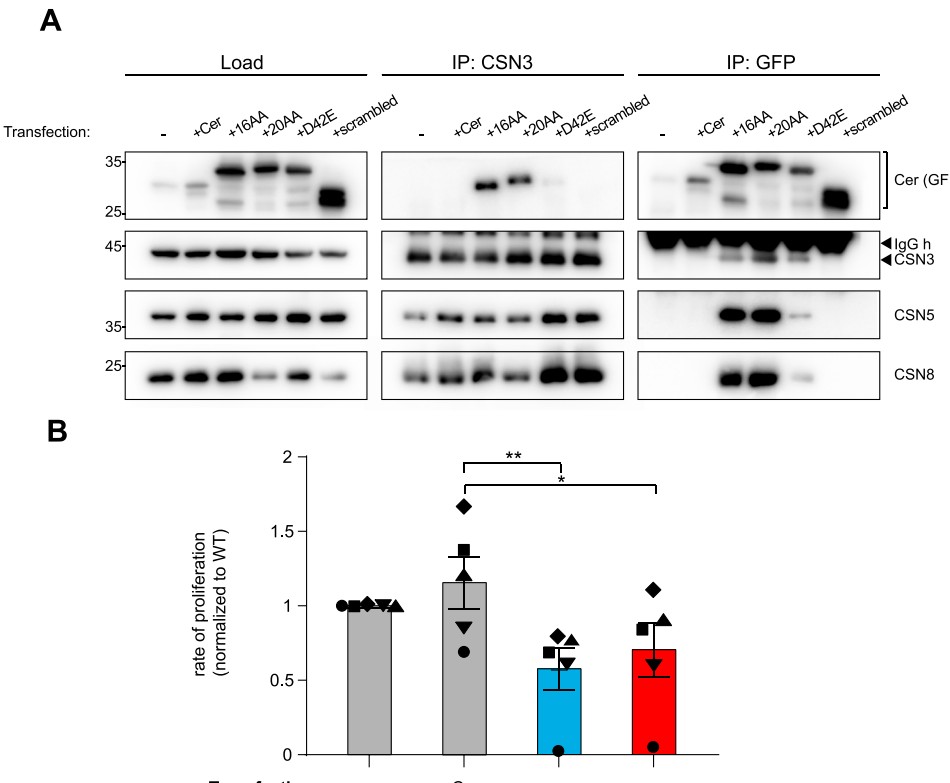

**B**

**Figure 6. Expression of C-CSNAP reduces cellular proliferation rate.**
**(A)** Immunoprecipitation analysis using anti-GFP antibody recognizing cerulean (Cer) from lysates of cells expressing the Cer-fused 16AA, 20AA, D42E, and scrambled C-CSNAP in HAP1 WT cells. The COP9 signalosome complex was pulled down from all cells, except when Cer is expressed alone or fused to the scrambled C-CSNAP. Similarly, reciprocal immunoprecipitation by an anti-CSN3 antibody pulled down only the Cer-fused 16AA, 20AA, or D42E C-CSNAP. **(B)** Stable overexpression of 16AA and 20AA C-CSNAP–Cer in HAP1 WT cells reduces their rate of proliferation. The bar plot shows average proliferation rates of each cell line using five independent replicates with SEM. One-way ANOVA ($P < 0.05$) with Dunnett's multiple comparisons test was used to compare means *$P < 0.05$, **$P < 0.01$.

measurment one-way ANOVA ($P < 0.05$) with Dunnett's multiple comparisons test was used to compare means to Cer as control *$P < 0.05$, **$P < 0.01$.

# Supplementary Information

# Acknowledgements

M Sharon is grateful for the support of the Sagol Institute for Longevity Research grant and a Moross Proof-of-Concept grant. M Sharon is the incumbent of the Aharon and Ephraim Katzir Memorial Professorial Chair. S Fleishman was supported by the Dr. Barry Sherman Institute for Medicinal Chemistry.

## Author Contributions

MG Füzesi-Levi: conceptualization, data curation, formal analysis, and writing—original draft.
G Ben-Nissan: conceptualization and data curation.
D Listov: data curation and formal analysis.
Y Fridmann Sirkis: data curation.
Z Hayouka and S Fleishman: conceptualization.
M Sharon: conceptualization, methodology, and writing—original draft, review, and editing.

## Conflict of Interest Statement

The authors declare that they have no conflict of interest.

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
