## [Reviewer comments · Life Science Alliance]

Life Science Alliance

The C-terminal tail of CSNAP attenuates the CSN complex

Maria Füzési-Levi, Gili Ben-Nissan, Dina Listov, Yael Friedmann Sirkis, Zvi Hayouka, Sarel Fleishman, and Michal Sharon

DOI: <https://doi.org/10.26508/lsa.202201634>

Corresponding author(s): Michal Sharon, Weizmann Institute of Science

Review Timeline:

Submission Date:	2022-07-27
Editorial Decision:	2022-09-09
Revision Received:	2023-06-04
Editorial Decision:	2023-06-23
Revision Received:	2023-06-29
Accepted:	2023-07-03

Scientific Editor: Novella Guidi

Transaction Report:

September 9, 2022

Re: Life Science Alliance manuscript #LSA-2022-01634-T

Prof. Michal Sharon
Weizmann Institute of Science
Department of Biological Chemistry
2 Herzl St.
Rehovot, Select State 76100
Israel

Dear Dr. Sharon,

Thank you for submitting your manuscript entitled "The C-terminal tail of CSNAP attenuates the CSN complex" to Life Science Alliance. The manuscript was assessed by expert reviewers, whose comments are appended to this letter. We invite you to submit a revised manuscript addressing the Reviewer comments.

Thank you for this interesting contribution to Life Science Alliance. We are looking forward to receiving your revised manuscript.

Sincerely,

B. MANUSCRIPT ORGANIZATION AND FORMATTING:

Reviewer #1 (Comments to the Authors (Required)):

The CSN is a multi-protein complex that regulates ubiquitin-mediated protein degradation. Specifically, this regulation is mediated by (i) its activity as a deneddylase, removing the ubiquitin homolog Nedd8 from its conjugation to E3 ligase subunits (CRL subfamily) and (ii) "locking up" the cognate CRL by complexation, thereby preventing ubiquitin ligase activity. The CSN was initially characterized as a 8 subunit complex. Subsequently, the Sharon group identified a mini-protein (CSNAP) that integrated into the complex, regulating activity. In a follow up study, the Sharon group reported that CSNAP impacts on normal CSN biological activity such that cells lacking CSNAP experience deleterious effects on cell cycle progression and viability.

The major aim of the present study is to characterize the molecular interaction of CSNAP with CSN and design peptides that can compete with endogenous CSNAP, rendering the CSN less active. In effect, these peptides would engender results similar to delta CSNAP. This in turn would be a starting point for pharmacological targeting of CSN, a strategy pursued previously for cancer treatment by drugging the enzymatic subunit, CSN5. The authors have thus found that the last 16-20 residues of CSNAP are necessary and sufficient for incorporation into CSN. To this end, they modeled the interaction using AlphaFold, checked it using constraints from chemical crosslinking results previously published and then further characterized it by immunoprecipitation assays, and peptide arrays. Peptide validation was pursued by mass spectrometry of in vitro samples. Finally, they showed results that the candidate peptides (actually, in the form of fusion to fluorescent protein (Cerulean)) had an effect on cell viability.

This study is meant as a proof of concept for pharmacologically targeting CSN. It provides results that are novel although it is not clear if we can conclude with a really high degree of confidence that these peptides have "the potential to be further developed as lead compounds opening a new avenue for selective CSN inhibition." I leave it to the editors to assess whether this type of study with a translational bent is appropriate for the LSA. As for the molecular biology that can be gleaned, we have learned more about CSNAP interacts with CSN but very little about how mechanistically CSNAP functions. As for the binding characterization, it would be nice if an in vitro system had been established enabling quantitative characterization of CSNAP and the peptides' association with CSN (affinity measurements). As for proof of concept, it would be nice if the designed peptides could be shown to modulate CSN function per se and not just cell proliferation, ensuring that the proposed "drug" mechanism was indeed mediated by an "attack" on CSN. As a follow on to that point, the cell proliferation results employed in fig. 5 might be more convincing if a scrambled peptide (fused to the Cerulean) and a potent mutant (from fig. 3) were used as controls. I list below suggestions for changes to text or figures.

p 4: "showing remarkable similarity." I fail to see how this is remarkable. The structure is in the PDB so there is an exact (100%) template for it.

Figure 1:

panel B: It is very difficult to follow this graphic. Suggested changes: inset of CSN cartoon, with CSN1-3-8 highlighted. Use of a transparent surface, with an underlying secondary structure cartoon depiction. This will also facilitate visualization of the CSN helical bundle sub-domain mentioned on p 6.

panels C and B. It would helpful to show the residues on CSN3 which are listed on p 6 (Glu 133 and 333).

panel D: A label for the CSNAP CT would be helpful to orient the reader.

p 6: "Cross-links involving the C-terminus of CSN1 (positions 472 and 477) were not considered, due to the multiple degrees of freedom of this region." This passage needs to be clarified. Otherwise, it seems as if the authors are cherry-picking the constraints.

Figure 2:

I find that the peptide sequences in panel A are difficult to read. Not enough contrast. Also, the peptides should have labels. For example, not clear to me what is the peptide sequence for 46D.

Figure 3:

panel B: This panel was very difficult to decipher. There is no point of listing the actual peptide sequence since it is not legible. Best to put it into a table in supplementary. Histogram (x-axis) should be the number of peptide residues. There is no explicit

description/meaning of red, blue, grey bars listed in the legend.

panel C: Again, the peptide sequence here is not useful. Better to have the sequence listed above the histogram bars with residue position numbers below or above each amino acid as an inset and then histogram bars numbered accordingly.

panels E-F-G: The peptide sequences are not legible. Best to list the residue number of the mutation for each bar.

p 9: "the binding suggesting that the larger solvent exposed increase polar surface of glutamic acid ..." The word increase should be deleted.

Figure 4: There is no explicit explanation for the different subunit coloring in the cartoon. A color legend would be useful. In the legend, the second (B) should be deleted.

p 13: "The C-terminal region of CSNAP, however, remains disordered in its CSN-bound state," This reviewer finds this terminology quite misleading. How can it be disordered and yet bound? The authors even have modeled a conformation (Fig. 1A). Yes, it is in an extended conformation which does not fit into standard secondary structure parameters but if it binds it cannot be disordered. In my mind, disordered implies a completely dynamic structure without any favored conformations. Clearly, there is a favored conformation enabling the interaction. I appreciate that it may be a rapid and short-lived interaction, although no kinetic studies are available so we really do not know, but it certainly is enough to enable function.

Reviewer #2 (Comments to the Authors (Required)):

This work describes the identification and characterization of C-terminal peptides of CSNAP to understand its potential role in modulating CSN activity. First, the authors examined the interaction of the C-terminus of CSNAP with CSN 1, 3 and 8 subunits using alpha-fold prediction, which correlated well with existing structural data. Next, the authors performed IP-western experiments to identify a 16-AA C-terminal peptide with highest binding affinity to the CSN complex. Extensive peptide array experiments were carried out to further explore the binding of C-peptides to CSN. In combination with Native MS analysis, it was suggested that C-peptides could displace endogenous CSNAP in the CSN complex. In addition, overexpression of C-peptides appears to reduce cell proliferation, based on which the authors concluded that C-peptides of CSNAP can inhibit CSN activity and has a potential in future therapeutics targeting CSN. While the content is scientifically interesting, additional evidence would be needed to support the conclusions especially on the displacement of endogenous CSNAP and the inhibitory effect of CSN by C-peptides of CSNAP. Here are the specific points that need clarification:

1. Figure 2 illustrates the co-purification of CSN5 with C-terminal peptides of CSNAP. It is surprising that the 42-57 peptide would have better binding to CSN5 than the full-length CSNAP. As the expression levels of different constructs could impact their pulldowns, it would be important to include the blots showing protein levels in lysates. To show whether the CSN complex was co-purified, it would be important to probe additional CSN subunits. In addition, the observed results could be influenced by protein tagging and overexpression. It is unclear whether the Cer tag was fused at the N-terminus or C-terminus of C-peptides and full-length CSNAP. It would be helpful to show that protein tagging does not interfere with CSN interactions. Moreover, reciprocal co-IP using another CSN subunit as the bait would be helpful to confirm the incorporation of C-peptides of CSNAP into native CSN in cells.
2. Based on the native MS, it was suggested that CSNAP C-peptides could displace endogenous CSNAP. However, the presence and distribution of intact CSN1-8, CSN1-9 and CSN1-8+C-peptides under different conditions was not clearly presented. As CSNAP is not required for the assembly and function of canonical CSN complex, CSN1-8 and CSN1-9 can both exist in cells. Based on the current experimental design, the detected C-peptides in the mixture could be due to their binding to the CSN1-8 and/or the CSN1-9 complexes. Therefore, it is unclear whether C-peptides could indeed displace CSNAP in vitro and in vivo. Additional evidence would be needed to support the claim.
3. To confirm the effects of C-peptides on the incorporation of CSNAP to CSN complex, both C-peptides and full-length CSNAP need to be probed in CSN co-IPed samples. In addition, CSN subunits need to be probed in full-length CSNAP pulldowns in the presence and absence of C-peptides.
4. It is unclear how the authors could determine the inhibitory effect of CSNAP C-peptides on the CSN complex based on the reduced cell proliferation revealed by resazurin-based viability assays. To illustrate the importance of CSNAP C-peptides, its impact on CSN deneddylase activity needs to be demonstrated in vitro and in cells. The effects of C-peptides should be compared with that of full-length CSNAP.
5. Please check text errors. For example, Fig 5B shows cell proliferation data, not Fig 5A. It was stated that "The data show that there are significant reductions in cell proliferation in both 16AA and 20AA C-CSNAP clones compared to the WT cells or cells expressing full-length CSNAP fused to Cer cells (Fig. 5A)." (page 12)

Reviewer #1

The CSN is a multi-protein complex that regulates ubiquitin-mediated protein degradation. Specifically, this regulation is mediated by (i) its activity as a deneddylase, removing the ubiquitin homolog Nedd8 from its conjugation to E3 ligase subunits (CRL subfamily) and (ii) "locking up" the cognate CRL by complexation, thereby preventing ubiquitin ligase activity. The CSN was initially characterized as a 8 subunit complex. Subsequently, the Sharon group identified a mini-protein (CSNAP) that integrated into the complex, regulating activity. In a follow up study, the Sharon group reported that CSNAP impacts on normal CSN biological activity such that cells lacking CSNAP experience deleterious effects on cell cycle progression and viability.

The major aim of the present study is to characterize the molecular interaction of CSNAP with CSN and design peptides that can compete with endogenous CSNAP, rendering the CSN less active. In effect, these peptides would engender results similar to delta CSNAP. This in turn would be a starting point for pharmacological targeting of CSN, a strategy pursued previously for cancer treatment by drugging the enzymatic subunit, CSN5. The authors have thus found that the last 16-20 residues of CSNAP are necessary and sufficient for incorporation into CSN. To this end, they modeled the interaction using AlphaFold, checked it using constraints from chemical crosslinking results previously published and then further characterized it by immunoprecipitation assays, and peptide arrays. Peptide validation was pursued by mass spectrometry of in vitro samples. Finally, they showed results that the candidate peptides (actually, in the form of fusion to fluorescent protein (Cerulean)) had an effect on cell viability.

This study is meant as a proof of concept for pharmacologically targeting CSN. It provides results that are novel although it is not clear if we can conclude with a really high degree of confidence that these peptides have "the potential to be further developed as lead compounds opening a new avenue for selective CSN inhibition." I leave it to the editors to assess whether this type of study with a translational bent is appropriate for the LSA.

We thank the reviewer for this positive comment, indicating that the study is novel. We have carefully studied your comments and suggestions, and addressed all of them, as described below.

1. As for the molecular biology that can be gleaned, we have learned more about CSNAP interacts with CSN but very little about how mechanistically CSNAP functions.

We accept the reviewer's comment and agree that additional information should have been provided. We have now revised the introduction to indicate that we have previously shown that CSNAP reduces the affinity of CSN toward CRL complexes leading to global effects on CRLs and that removing CSNAP does not affect the deneddylation activity (Füzesi-Levi *et al.* 2020, *Cell Death Differ* 27:984–998).

2. As for the binding characterization, it would be nice if an in vitro system had been established enabling quantitative characterization of CSNAP and the peptides' association with CSN (affinity measurements).

We thank the reviewer for this suggestion and have performed quantitative analysis of protein-peptide interactions using Biacore assay (surface plasmon resonance biosensor assay) (new Figure 4). These experiments indicate that the 16 and 20 amino acid peptides bind the CSN^{ΔCSNAP} complex with a similar

K_d of 4.0×10^{-10} and 4.3×10^{-10} , respectively. The D42E peptide displayed a slightly lower affinity of 2.5×10^{-9} and the highest affinity was measured for the full-length CSNAP protein, 6.8×10^{-11} . No measurable binding was detected for the scrambled peptide. Overall, these experiments support the peptide array and mass spectrometry experiments, indicating the specificity and high affinity of the C-CSNAP peptides to CSN.

3. As for proof of concept, it would be nice if the designed peptides could be shown to modulate CSN function per se and not just cell proliferation, ensuring that the proposed "drug" mechanism was indeed mediated by an "attack" on CSN. As a follow on to that point, the cell proliferation results employed in fig. 5 might be more convincing if a scrambled peptide (fused to the Cerulean) and a potent mutant (from fig. 3) were used as controls.

We agree with the reviewer that it would be valuable to use a scrambled peptide to confirm the specificity of the C-CSNAP peptide variants. We therefore performed Western blot and Biacore assays with a scrambled peptide (new Figures 4, 6A and Supplementary Figure 5). In these experiments, no detectable interaction between the scrambled peptide and the CSN complex was observed. In view of these results, for the cellular assays, we decided not to fuse the scrambled peptide to cerulean, as it will not provide additional input, rather it will act as the cerulean negative control.

4. p 4: "showing remarkable similarity." I fail to see how this is remarkable. The structure is in the PDB so there is an exact (100%) template for it.

AlphaFold is not a homology-modeling technique and while it may have memorized certain structural aspects learned from the PDB, it is highly unlikely that it memorized the structure of such a large complex. Moreover, we did not run AlphaFold using templates, the model coordinates are not identical to those seen in the PDB and no structure in the PDB contains the CSNAP peptide. Nevertheless, in the revised version of the manuscript, we have tuned down the phrasing and clarified that the model structure is an AlphaFold prediction.

5. Figure 1: panel B: It is very difficult to follow this graphic. Suggested changes: inset of CSN cartoon, with CSN1-3-8 highlighted. Use of a transparent surface, with an underlying secondary structure cartoon depiction. This will also facilitate visualization of the CSN helical bundle sub-domain mentioned on p 6. Panels C and B. It would helpful to show the residues on CSN3 which are listed on p 6 (Glu 133 and 333). Panel D: A label for the CSNAP CT would be helpful to orient the reader.

We agree with the reviewer and have corrected the Figure accordingly (new Figure 1).

6. p 6: "Cross-links involving the C-terminus of CSN1 (positions 472 and 477) were not considered, due to the multiple degrees of freedom of this region." This passage needs to be clarified. Otherwise, it seems as if the authors are cherry-picking the constraints.

Following the referee advice, we have rephrased the sentence.

7. Figure 2: I find that the peptide sequences in panel A are difficult to read. Not enough contrast. Also, the peptides should have labels. For example, not clear to me what is the peptide sequence for 46D.

We thank the reviewer for this comment and have corrected the Figure accordingly. Moreover, to make the figure clearer we divided it to two separate figures. In the main text the immunoprecipitation results of WT cells are shown (new Fig. 2), and in the supplementary information (new Supplementary Figure 2), we display the data for cells lacking CSNAP (CSN^{ΔCSNAP}).

8. Figure 3: panel B: This panel was very difficult to decipher. There is no point of listing the actual peptide sequence since it is not legible. Best to put it into a table in supplementary. Histogram (x-axis) should be the number of peptide residues. There is no explicit description/meaning of red, blue, grey bars listed in the legend.

We thank the reviewer for these suggestions and have corrected both Figure 3 and Supplementary Figure 3, accordingly. We have also added Supplementary Tables 1 and 2, which include the list of the peptides used in each panel and their sequence for the CSN and CSN^{ΔCSNAP} complexes, respectively. In addition, we corrected the legends to include the description of the red, blue and gray bars.

9. panel C: Again, the peptide sequence here is not useful. Better to have the sequence listed above the histogram bars with residue position numbers below or above each amino acid as an inset and then histogram bars numbered accordingly. Panels E-F-G: The peptide sequences are not legible. Best to list the residue number of the mutation for each bar.

We agree with the reviewer and in panels C-F we have listed only the number of the mutation for each bar. The full sequences of the different peptides are detailed in new Supplementary Tables 1 and 2.

10. p 9: "the binding suggesting that the larger solvent exposed increase polar surface of glutamic acid ..." The word increase should be deleted.

We thank the reviewer for spotting this mistake, and have corrected it.

11. Figure 4: There is no explicit explanation for the different subunit coloring in the cartoon. A color legend would be useful. In the legend, the second (B) should be deleted.

We thank the reviewer for this comment and have corrected the text accordingly.

12. p 13: "The C-terminal region of CSNAP, however, remains disordered in its CSN-bound state," This reviewer finds this terminology quite misleading. How can it be disordered and yet bound? The authors even have modeled a conformation (Fig. 1A). Yes, it is in an extended conformation which does not fit into standard secondary structure parameters but if it binds it cannot be disordered. In my mind, disordered implies a completely dynamic structure without any favored conformations. Clearly, there is a favored conformation enabling the interaction. I appreciate that it may be a rapid and short-lived interaction, although no kinetic studies are available so we really do not know, but it certainly is enough to enable function.

We understand the reviewer's comment and have modified the text accordingly.

This work describes the identification and characterization of C-terminal peptides of CSNAP to understand its potential role in modulating CSN activity. First, the authors examined the interaction of the C-terminus of CSNAP with CSN 1, 3 and 8 subunits using alpha-fold prediction, which correlated well with existing structural data. Next, the authors performed IP-western experiments to identify a 16-AA C-terminal peptide with highest binding affinity to the CSN complex. Extensive peptide array experiments were carried out to further explore the binding of C-peptides to CSN. In combination with Native MS analysis, it was suggested that C-peptides could displace endogenous CSNAP in the CSN complex. In addition, overexpression of C-peptides appears to reduce cell proliferation, based on which the authors concluded that C-peptides of CSNAP can inhibit CSN activity and has a potential in future therapeutics targeting CSN. While the content is scientifically interesting, additional evidence would be needed to support the conclusions especially on the displacement of endogenous CSNAP and the inhibitory effect of CSN by C-peptides of CSNAP. Here are the specific points that need clarification:

We thank the reviewer for the general positive opinion and for commenting that this study is scientifically interesting. We have carefully studied the reviewers' comments and suggestions, and addressed all of them, as described below. We added new data, and refined the text, as suggested.

1. Figure 2 illustrates the co-purification of CSN5 with C-terminal peptides of CSNAP. It is surprising that the 42-57 peptide would have better binding to CSN5 than the full-length CSNAP. As the expression levels of different constructs could impact their pulldowns, it would be important to include the blots showing protein levels in lysates. To show whether the CSN complex was co-purified, it would be important to probe additional CSN subunits. In addition, the observed results could be influenced by protein tagging and overexpression. It is unclear whether the Cer tag was fused at the N-terminus or C-terminus of C-peptides and full-length CSNAP. It would be helpful to show that protein tagging does not interfere with CSN interactions. Moreover, reciprocal co-IP using another CSN subunit as the bait would be helpful to confirm the incorporation of C-peptides of CSNAP into native CSN in cells.

We agree with the reviewer that blots showing the protein levels in the lysates should have been added. In the revised version of the manuscript, we have included this information in new Figure 2B and Supplementary Figure 2B.

As for N- or C- terminal tagging of CSNAP, we understand the reviewers concern and would like to indicate that previously we have examined this aspect by fusing cerulean (Cer) to either terminus of CSNAP and performing reciprocal co-immunoprecipitation analyses, using antibodies against CSN3 and GFP (Rozen *et al.* 2015, *Cell Rep*, 13(3):585). The results indicated that fusing Cer to either terminus of CSNAP did not hinder its ability to interact with CSN (see Figure below).

Cellular proteins extracted from the different fluorescently tagged HEK293 cell lines were immunoprecipitated, using anti-GFP and anti-CSN3 antibodies. As control, lysate from HEK293 cells stably expressing FLAG-CSN2 was used. Lysates (L) were run side by side with their corresponding immunoprecipitated proteins (IP) and visualized using various antibodies, as indicated (IB). Results show that fusing Cer to either terminus of CSNAP did not hinder its ability to interact with CSN.

In response to the comment regarding the reciprocal co-immunoprecipitation analyses, as suggested by the reviewer we performed experiments using various other CSN subunit as the bait (not CSN3). However regardless of the antibody we used we could not pull-down CSN, even with the positive control. Thus, unlike the anti-CSN3 antibody, our antibodies against CSN5, CSN8 and CSN1, were not suitable for immunoprecipitation analyses. Nevertheless, we repeated the co-immunoprecipitation analyses and probed the Western blots not only for CSN3 also but also for CSN5 and CSN8 (new Figure 6A).

2. Based on the native MS, it was suggested that CSNAP C-peptides could displace endogenous CSNAP. However, the presence and distribution of intact CSN1-8, CSN1-9 and CSN1-8+C-peptides under different conditions was not clearly presented. As CSNAP is not required for the assembly and function of canonical CSN complex, CSN1-8 and CSN1-9 can both exist in cells. Based on the current experimental design, the detected C-peptides in the mixture could be due to their binding to the CSN1-8 and/or the CSN1-9 complexes. Therefore, it is unclear whether C-peptides could indeed displace CSNAP in vitro and in vivo. Additional evidence would be needed to support the claim.

We accept the reviewer's comment and agree that additional information should have been provided to emphasize the fact that in cells only the 9 subunit CSN complex exists (CSN1-9). This has been demonstrated by us in our previous study, which indicated that CSNAP is an integral and stoichiometric subunit of the CSN, present in equimolar amounts like the other CSN subunits (Rozen *et al.* 2015, *Cell Rep*, 13(3):585). We revised the manuscript to clarify this point.

3. To confirm the effects of C-peptides on the incorporation of CSNAP to CSN complex, both C-peptides and full-length CSNAP need to be probed in CSN co-IPed samples. In addition, CSN subunits need to be probed in full-length CSNAP pulldowns in the presence and absence of C-peptides.

Following the Reviewer's suggestion, we have added this data (new Figure 6A), which demonstrates that the 16 and 20 amino acid C-CSNAP peptides as well as the D42E C-CSNAP peptide are incorporated with the CSN complex. Cerulean and the scrambled peptide served as negative control. In this experiment, in addition to an anti-CSN3 antibody, antibodies against CSN5 and CSN8 were used for probing the immunoprecipitated samples. We would also like to state that in the C-CSNAP peptide displacement analysis of full-length peptides, following streptactin bead isolation of CSN, confirmed the presence of the C-CSNAP peptides and CSN subunits (Figure 5 and Supplementary Figure 4).

4. It is unclear how the authors could determine the inhibitory effect of CSNAP C-peptides on the CSN complex based on the reduced cell proliferation revealed by resazurin-based viability assays. To illustrate the importance of CSNAP C-peptides, its impact on CSN deneddylase activity needs to be demonstrated in vitro and in cells. The effects of C-peptides should be compared with that of full-length CSNAP.

In our previous studies, we have shown that CSNAP does not significantly affect the catalytic deneddylation capacity of the CSN complex (Rozen *et al.* 2015, *Cell Rep*, 13(3):585 and Füzesi-Levi *et al.* 2020, *Cell Death Differ* 27:984–998). In these studies, we have demonstrated that WT and Δ CSNAP cells exhibit a similar rate of deneddylation. This result is in accordance with other studies that compared the rate of deneddylation of endogenous CSN prepared from HEK293 cells, with that of recombinant CSN lacking CSNAP (CSN ^{Δ CSNAP}) (Enchev *et al.* 2012, *Cell Rep*, 2:616–27 and Emberley *et al.* 2012, *J Biol Chem*, 287:29679–89). Therefore, we think that reproducing these results again will not add to the novelty of the manuscript. Nevertheless, in response to this comment we have modified the text to emphasize the fact that CSNAP does not affect the deneddylation activity.

5. Please check text errors. For example, Fig 5B shows cell proliferation data, not Fig 5A. It was stated that "The data show that there are significant reductions in cell proliferation in both 16AA and 20AA C-CSNAP clones compared to the WT cells or cells expressing full-length CSNAP fused to Cer cells (Fig. 5A)." (page 12)

We apologize for this oversight and have corrected the text accordingly.

June 23, 2023

RE: Life Science Alliance Manuscript #LSA-2022-01634-TR

Prof. Michal Sharon
Weizmann Institute of Science
Department of Biological Chemistry
2 Herzl St.
Rehovot, Select State 76100
Israel

Dear Dr. Sharon,

Thank you for submitting your revised manuscript entitled "The C-terminal tail of CSNAP attenuates the CSN complex". We would be happy to publish your paper in Life Science Alliance pending final revisions necessary to meet our formatting guidelines.

- please clarify the final Reviewer 2's concern
- please upload your Tables in editable .doc or Excel format
- please add the Twitter handle of your host institute/organization as well as your own or/and one of the authors in our system
- please remove your figures from the manuscript text. All figure files should be uploaded as individual ones, including the supplementary figure files; all figure legends should only appear in the main manuscript file
- please add your main, supplementary figure, and table legends to the main manuscript text after the references section;
- please add an Author Contributions section to your main manuscript text
- please add a conflict of interest statement to your main manuscript text
- Table should include a brief descriptive title and be labeled as Table 1
- please add callouts for Figures 4A-c and 4D-F; 5A, B, D; S1A; S2A; S3A-D to your main manuscript text;

A. FINAL FILES:

B. MANUSCRIPT ORGANIZATION AND FORMATTING:

Sincerely,

Reviewer #2 (Comments to the Authors (Required)):

While the authors have adequately addressed most of the previous concerns, one remaining question is about the inhibitory effect of CSNAP C-peptides on the CSN complex, which was described as the main finding of the paper. However, there is no direct evidence to support such conclusion. As the authors stated in the rebuttal letter that CSNAP does not significantly affect the known function of the CSN complex, namely its deneddylase activity (Rozen et al. 2015, Cell Rep, 13(3):585 and Fűzesi-Levi et al. 2020, Cell Death Differ 27:984-998), it is anticipated that CSNAP C-peptides would not impact CSN deneddylase activity. Thus, it is unclear how CSNAP C-peptides would attenuate CSN activity and what activity of CSN is inhibited by CSNAP C-peptides to have therapeutic potential. Please clarify.

July 3, 2023

RE: Life Science Alliance Manuscript #LSA-2022-01634-TRR

Prof. Michal Sharon
Weizmann Institute of Science
Department of Biological Chemistry
234 Herzl St.
Rehovot, Select State 76100
Israel

Dear Dr. Sharon,

Thank you for submitting your Research Article entitled "The C-terminal tail of CSNAP attenuates the CSN complex". It is a pleasure to let you know that your manuscript is now accepted for publication in Life Science Alliance. Congratulations on this interesting work.

DISTRIBUTION OF MATERIALS:

Again, congratulations on a very nice paper. I hope you found the review process to be constructive and are pleased with how the manuscript was handled editorially. We look forward to future exciting submissions from your lab.

Sincerely,
